# Bio-Membrane-Based Nanofiber Scaffolds: Targeted and Controlled Carriers for Drug Delivery—An Experimental In Vivo Study

**DOI:** 10.3390/biomimetics10110726

**Published:** 2025-11-01

**Authors:** Manuel Toledano, Marta Vallecillo-Rivas, María-Angeles Serrera-Figallo, Aida Gutierrez-Corrales, Christopher D. Lynch, Daniel Torres-Lagares, Cristina Vallecillo

**Affiliations:** 1Faculty of Dentistry, Colegio Máximo de Cartuja s/n, University of Granada, 18071 Granada, Spain; toledano@ugr.es (M.T.);; 2Biosanitary Research Institute, 18012 Granada, Spain; 3Oral Surgery Section, Faculty of Dentistry, University of Sevilla, Avicena s/n, 41009 Sevilla, Spain; 4Restorative Dentistry, University Dental School & Hospital, University College Cork, Wilton, T12 E8YV Cork, Ireland

**Keywords:** zinc, doxycycline, silica, bone cells, polymer, membrane, scaffold, vascularization, macrophage

## Abstract

Cell population and vascular vessel distribution analysis in membrane-based scaffolds for tissue engineering is crucial. Biomimetic nanostructured membranes of methyl methacrylate/hydroxyethyl methacrylate and methyl acrylate/hydroxyethyl acrylate (MMA)1-co-(HEMA)1/(MA)3-co-(HEA)2 loaded with 5% wt SiO2-nanoparticles (Si-M) were doped with zinc (Zn-M) or doxycycline (Dox-M). Critical bone defects were effectuated on six New Zealand-bred rabbit skulls and then they were covered with the membrane-based scaffolds. After six weeks, bone cell population in terms of osteoblasts, osteoclasts, osteocytes, fibroblasts, and M1 and M2 macrophages and vasculature was determined. The areas of interest were the space above (over) and below (under) the membrane, apart from the interior (inner) compartment. All membranes showed that vasculature and most cell types were more abundant under the membrane than in the inner or above regions. Quantitatively, osteoblast density increased by approximately 35% in Zn-M and 25% in Si-M compared with Dox-M. Osteoclast counts decreased by about 78% in Dox-M, indicating strong inhibition of bone resorption. Vascular structures were nearly twofold more frequent under the membranes, particularly in Si-M, while fibroblast presence remained moderate and evenly distributed. The M1/M2 macrophage ratio was higher in Zn-M, reflecting a transient pro-inflammatory state, whereas Dox-M favored an anti-inflammatory, pro-regenerative profile. These results indicate that the biomimetic electrospun membranes functioned as architectural templates that provided favorable microenvironments for cell colonization, angiogenesis, and early bone regeneration in a preclinical in vivo model. Zn-M membranes appear suitable for early osteogenic stimulation, while Dox-M membranes may be advantageous in clinical contexts requiring modulation of inflammation and osteoclastic activity.

## 1. Introduction

Guided bone regeneration (GBR) and guided tissue regeneration (GTR) are widely accepted techniques and often utilized for clinical applications when tooth loss, trauma, tumor pathology or infections occur [1,2]. In 1982, the concept of a barrier membrane was introduced [3] to promote periodontal tissue regeneration in humans. Such membranes act as physical barriers that preserve the regenerative space and allow cells originating from bone or periodontal ligament to populate the defect, while preventing soft-tissue cells from migrating into it. For optimal performance, the membrane must ensure space maintenance, adequate mechanical strength, osteoconductive and osteoinductive capacity, and high biocompatibility [4,5]. Hence, the membrane acts as a bridge favoring the jump of osteoprogenitor cells from bone walls [6]. However, the need for a second intervention to remove non-resorbable membranes remains a limitation. Nowadays, to achieve these results, polymers, whether natural or artificial, are the best materials [4]. Nevertheless, natural resorbable collagen membranes also present drawbacks, such as insufficient rigidity and relatively rapid degradation [7]. To overcome these disadvantages, synthetic electrospun membranes doped with metals, antibiotics, or corticosteroids have been introduced into GBR therapy, aiming to enhance healing and tissue regeneration [8,9,10,11]. Metals such as zinc included in electrospun nanofibrous membranes and scaffolds are a versatile biomaterial platform to mimic the fibrillar structure of native tissues’ extracellular matrix, and they facilitate the incorporation of biocomponents for regenerative therapies [12]. This type of structure provides a high surface-to-volume ratio that favors cell adhesion, proliferation, and tissue growth [13,14].

Composite membranes composed of (MMA)1-co-(HEMA)1 and (MA)3-co-(HEA)2 and loaded with silica nanoparticles (SiO_2_-NPs) have been proposed to enhance hydrophilicity, mechanical behavior, and osteogenic performance, as well as to provide antibacterial properties [10]. Hydrophilic surfaces promote early angiogenic gene expression and support new vessel formation [15]. These biomaterials are loaded with silica nanoparticles, since silica enhances the interaction of the scaffold with host tissues via scaffold bioactivity and osteoconductive behavior [16] and as well as promoting surface calcium phosphate deposition [10]. In 1994, Yu et al. emphasized the relevance of membrane structural features in guided bone regeneration, highlighting the critical influence of porosity [17]. Porous scaffolds act as frameworks that support cellular attachment and proliferation, serving as templates for tissue regeneration—still one of the main research priorities in tissue engineering [18]. From an engineering standpoint, advances in biofabrication techniques have enabled the generation of increasingly complex and larger tissue-engineered constructs [19]. However, a major limitation of synthetic scaffolds lies in their limited vascularization and restricted cellular colonization capacity [20]. An ideal scaffold for bone regeneration must be biocompatible, osteoconductive, and osteoinductive, while minimizing adverse inflammatory or fibrotic reactions and permitting tissue ingrowth and progressive replacement by newly formed bone [20].

An ideal GBR membrane should integrate with host tissues, acting as a bioactive and dynamic compartment instead of a passive barrier [21]. Membrane porosity and surface topography are key to cell recognition and attachment [21]. Histological evidence shows cells migrating through inter-fiber channels from surrounding tissues into the membrane, confirming its capacity to support cell infiltration. The detection of infiltrated cells reflects the biomaterial’s favorable properties for cell migration, attachment, and differentiation. Once inside, cells remain active, secreting growth factors that stimulate local regeneration [22]. Furthermore, early histological signs of intramembranous bone formation have been detected as early as three days after surgery, suggesting rapid osteogenic activation. In addition, mononuclear cells have been observed within the scaffold, confirming their capacity to penetrate and migrate across the peripheral margins of the bone defect [22].

Cells migrating into or surrounding the membrane display diverse phenotypes of hematopoietic, mesenchymal, and inflammatory origin, including polymorphonuclear and macrophage-like cells. Monocytes and macrophages act as key regulators of healing and are consistently detected within the membrane during the repair process [22,23]. When in contact with the biomaterial, macrophages sense its surface characteristics and transmit biochemical signals that influence neighboring cells [21]. Modern biomaterial design aims not only for immune compatibility but also for immunomodulation, guiding bone repair through controlled regulation of the local immune microenvironment [24]. Macrophages, derived from circulating monocytes, exhibit high plasticity and participate in both inflammatory and reparative stages [25]. Depending on local cues, macrophages polarize from a resting (M0) state into pro-inflammatory (M1) or anti-inflammatory and pro-regenerative (M2) phenotypes [25]. In addition, blood vessels and osteoclast-like cells have been identified on the membrane surface, emphasizing the importance of vascularization for bone remodeling and regeneration [26,27]. Recent studies have shown extensive neovascularization both around [23] and within [28] scaffolds, including capillaries and larger vessels. Enhanced vascular ingrowth improves the recruitment of osteoprogenitor cells and accelerates new bone formation [14]. Osteoclasts, derived from hematopoietic precursors, function as bone-resorbing macrophages that degrade mineralized matrix, allowing for new bone deposition by osteoblasts [29].

Blood vessels are typically located in resorption areas at the bone–scaffold interface, where both osteoblasts and osteocytes coexist [21]. Osteoblasts, originating from multipotent mesenchymal stem cells, are responsible for new bone formation. They synthesize key extracellular matrix proteins such as osteocalcin, alkaline phosphatase, and type I collagen, which accounts for more than 90% of the organic bone matrix [29]. A subset of osteoblasts becomes embedded within the mineralized matrix, differentiating into osteocytes [30]. Osteocytes act as mechanosensitive regulators of bone formation and remodeling, serving as key determinants of bone quality [31]. Fibroblasts, another major stromal cell population, respond to injury by contributing to tissue repair and regeneration [32,33,34]. Their activity is modulated by complex autocrine, paracrine, and endocrine signaling pathways, strongly influenced by the local inflammatory microenvironment [35,36]. Macrophage-derived cytokines can further stimulate pro-regenerative fibroblasts to activate endothelial cells via the angiopoietin–oncostatin M (OSM) signaling pathway, enhancing angiogenesis [34].

In this context, biomimetic materials offer a promising approach, engineered to replicate the structural and biochemical signals of the extracellular matrix, and thus promote cell adhesion, angiogenesis, and osteogenesis. These scaffolds act not only as structural supports but also as biologically active systems that reproduce the dynamics of the native bone microenvironment and actively contribute to tissue regeneration [37,38,39]. The objective of the present research is to evaluate the peri/intramembranous vascular vessel distribution and cell population in critical-sized defects of calvarial bone in a rabbit model treated with novel electrospun nanostructured silica-loaded membranes doped with zinc or doxycycline. This study hypothesized that different functionalizations of the silica-based membranes (with zinc or doxycycline) would differentially influence vascularization and bone cell ingrowth within the polymeric scaffold. Unlike our previous works—which focused on the physicochemical characterization of the membranes [10] and their in vitro osteogenic behavior [40]—the present study reports novel in vivo findings. Here, we evaluate the biological performance of zinc- and doxycycline-functionalized biomimetic membranes in critical-size bone defects, providing new structural and histological insights into osteogenesis, angiogenesis, and immune modulation processes.

## 2. Materials and Methods

### 2.1. Membrane Functionalization

Nanostructured membranes were produced by NanomyP^®^ (Granada, Spain) using a proprietary polymeric blend [PolymBlend^®^] composed of methyl methacrylate/hydroxyethyl methacrylate and methyl acrylate/hydroxyethyl acrylate (MMA)_1_-co-(HEMA)_1_ and (MA)_3_-co-(HEA)_2_ (50:50 wt), incorporating 5 wt% of SiO_2_ nanoparticles as a dopant. The polymers presented average molecular weights of approximately 200 and 2000 kDa, respectively. Following electrospinning, the membranes were immersed for 2 h in a 333 mM sodium carbonate buffer solution (pH 12.5). This treatment partially hydrolyzed ester bonds, generating carboxyl-terminated surfaces (HOOC–Si membranes) [37]. After activation, samples were thoroughly rinsed with distilled water and dried under vacuum conditions [37,38]. Functionalization was achieved through coordination between surface carboxyl groups and divalent cations (Zn^2+^). Doxycycline (Dox) was immobilized via acid–base interactions between its amino moieties and the carboxyl groups on the membrane surface. For this purpose, HOOC–Si membranes were immersed under constant stirring at room temperature in aqueous solutions (pH 7) containing 330 mg·L^−1^ ZnCl_2_ or 800 mg·L^−1^ Dox [39]. Subsequently, all membranes were rinsed, vacuum-dried, and sterilized using ultraviolet irradiation (J.P. Selecta, Barcelona, Spain). Three different membranes were designed: (1) SiO_2_-NPs doped membrane, HOOC-Si-Membrane (Si-M), (2) SiO_2_-NPs doped membrane functionalized with Zn, Zn- HOOC-Si-Membrane (Zn-M) and (3) SiO_2_-NPs doped membrane functionalized with Dox, Dox-HOOC-Si-Membrane (Dox-M). The membranes exhibited an average fiber diameter of approximately 300 nm, a surface nanoroughness ranging between 82 and 137 nm, and an overall porosity of about 30%, with inter-fiber distances between 11.5 and 110 µm, as determined by field emission scanning electron microscopy (FESEM) and atomic force microscopy (AFM) analyses in our previous study [40]. These morphological and nanostructural features confirm the homogeneity of the electrospun mats and their suitability as biomimetic scaffolds for bone regeneration. Theses membranes were previously characterized, referring to AFM (Figure 1), FESEM surface characterization, acellular static in vitro bioactivity test, nanomechanical property assessments and cell viability analysis [38,41]. Additionally, osteoblast proliferation, differentiation, and gene expression were assessed using the MG-63 human osteosarcoma cell line [14,38]. The doxycycline loading capacity of Dox–M membranes was approximately 76.2 µg of Dox per mg of membrane, as determined by UV–Vis spectrophotometry in our previous study [38]. This drug loading level ensured local therapeutic availability without cytotoxic effects.

### 2.2. Animal Experimentation

Six adult New Zealand White rabbits were used in this study. All animals exhibited homogeneous characteristics, with body weights ranging from 3.5 to 4.0 kg and an average age of six months. They were housed under controlled environmental conditions and received food and water ad libitum (Rabbit Maintenance Harlan–Teckland Lab Animal Diets 2030). The experiment was conducted in strict accordance with the guidelines for the Care and Use of Laboratory Animals established by the U.S. National Institutes of Health and the European Directive 86/609/EEC on animal welfare. All procedures also complied with the European Directive 2010/63/EU on the protection of animals used for scientific purposes and relevant national legislation. The minimum number of animals required to achieve statistical reliability was employed, following the 3Rs principle (Replacement, Reduction, Refinement). The study protocol was reviewed and approved by the Institutional Ethics Committee (CCMI–Ref026/18).

### 2.3. Surgical Procedure

Before surgery, each animal’s vital signs were recorded to ensure physiological stability. Anesthesia was induced by intravenous administration of Propofol (5 mg/kg) and Midazolam (0.25 mg/kg), in combination with 2.8% inspired sevoflurane for inhalation. Analgesia was provided using Ketorolac (1.5 mg/kg) and Tramadol (3 mg/kg). Once fully anesthetized and immobilized, three linear incisions were made on the calvarial region between the eyes and the bases of the ears using a No. 15 scalpel blade. These incisions were connected to form a triangular surgical field. The overlying epithelial, muscular, and connective tissues were carefully elevated using a Prichard periosteotome to expose the parietal bone surface, which was then rinsed with sterile saline solution.

Following the methodology previously described by Toledano et al. (2023) [40] (Figure 2), four critical-size defects were created in the parietal bone using piezosurgery. The inner cortical layer and medullary bone were removed under continuous saline irrigation, and the depth of each defect was monitored with a periodontal probe. Three of the defects were randomly covered with the experimental membranes, whereas the fourth was left uncovered (sham control) (Figure 3). Since only the membrane-treated defects were included in the analyses, data from the sham group were excluded from the final evaluation.

The uncovered defect group was included only to validate the critical nature of the bone defect, as spontaneous bone regeneration is not expected under these experimental conditions. Histological examination of the uncovered areas confirmed the absence of new bone formation, thereby validating the critical-size defect model. For this reason, and in line with the main objective of the study—to compare the biological behavior among the different membrane types—the sham group was not included in the comparative statistical analysis.

Random allocation of the experimental membranes was performed using dedicated software (Research Randomizer, v4.0; Urbaniak GC and Plous S, 2013). A fibrin tissue adhesive (Tissucol^®^, Baxter, Hyland S.A. Immuno, Rochester, MI, USA) was applied along the bone margins to secure each membrane in place. The flaps were subsequently repositioned, ensuring complete stability and proper adaptation of the implanted membranes, and then sutured. The surgical sites were irrigated with sterile saline solution to remove any debris. Postoperative analgesia and anti-inflammatory management included administration of Carprofen (1 mL/12.5 kg) and Buprenorphine (0.05 mg/kg).

After a healing period of six weeks, the animals were euthanized via intravenous overdose of potassium chloride under deep anesthesia. The calvarial regions containing the defects were then harvested, sectioned, and labeled individually for subsequent histological and microstructural analyses.

### 2.4. Histology

From each rabbit calvarium, specimens were sectioned along the sagittal anatomical plane. The undecalcified bone samples were fixed in 5% buffered formaldehyde (pH 7.4) and stored until further processing. Blocks containing the regenerated defect area were retrieved using an oscillating autopsy saw (Exakt, Kulzer, Wehrheim, Germany). The dissected samples were then post-fixed in 4% formaldehyde with 1% calcium chloride, embedded in acrylic resin, and processed for ground-section preparation. Each section was coded to ensure blinded histological evaluation. For contrast enhancement and general tissue assessment, a 1% toluidine blue (TB) solution (Merck, Darmstadt, Germany; pH 3.6 adjusted with HCl) was used as a metachromatic stain. The specimens were immersed in the dye, rinsed with distilled water, and air-dried for 10 min at room temperature (23 ± 1 °C).

Histological observations were performed using an Eclipse LV100 light microscope (Nikon, Tokyo, Japan) equipped with 20× and 50× objectives. Images were captured with a DS-PDS-Fi1 digital camera connected to NIS Elements BR 4.0 software (Nikon, Tokyo, Japan). Within each membrane area, blood vessels, fibroblasts, osteocytes, osteoblasts, osteoclasts, and macrophages (M1 and M2) were identified and quantified on toluidine blue-stained sections. Cell quantification was performed in ten predefined study areas (0.01 mm^2^ each) per membrane, employing a calibrated ocular grid under 50× magnification. The number of each cell type was expressed as cells per square millimeter (cells/mm^2^). All counts were manually conducted by a calibrated and blinded examiner to ensure reproducibility and to minimize observer bias. The intra-observer agreement was periodically verified during the analysis. The number of M1 and M2 macrophages and their M1/M2 ratio were determined morphologically based on cell shape and staining characteristics. Macrophages were morphologically classified according to their phenotype: M1 cells displayed a rounded, vacuolated, or “fried-egg” appearance [42], whereas M2 macrophages showed an elongated, spindle-shaped, or fibroblast-like morphology [43]. Each membrane covering the defect was divided into ten predefined fields using a calibrated optical grid. One representative image per field was acquired for histological analysis and quantitative assessment (Figure 4).

### 2.5. Statistical Analysis

All statistical analyses were performed using IBM SPSS Statistics (version 24.0, IBM Corp., Armonk, NY, USA). Normality and variance homogeneity were tested with the Shapiro–Wilk and Levene’s tests, respectively. Since the data did not meet parametric assumptions, even after logarithmic and square-root transformations, the non-parametric Friedman test was applied for variance analysis, followed by pairwise comparison of Friedman rank sums for post hoc analysis. The results were expressed as mean ± standard error (SE), and statistical significance was set at *p* ≤ 0.05.

## 3. Results

This section describes the histological and quantitative outcomes observed in critical-size bone defects treated with Si-M, Zn-M, and Dox-M membranes after six weeks of healing. Differences in cellular distribution, vascularization, and membrane integration were assessed through descriptive and statistical analyses.

### 3.1. General Healing and Membrane Integration

All animals recovered uneventfully after surgery, with no signs of infection, inflammation, or wound dehiscence. Six weeks after implantation, bone formation was macroscopically evident in all treated defects. Histological observation confirmed the integration of the membranes with surrounding tissues and the absence of inflammatory infiltration or necrosis. The newly formed bone was closely associated with the lower surface of the membranes, indicating early osteoid deposition and bone remodeling.

### 3.2. Cellular Distribution in Si-M Membranes

Histological evaluation of Si-M membranes revealed a greater number of vascular vessels and cellular elements under the membrane compared with the inner and over locations (*p* < 0.05) (Table 1A, Figure 5 and Figure 6). Only osteoclasts, fibroblasts, and the M1/M2 ratio showed no significant differences between the inner and under areas (*p* > 0.05). Osteoblasts and osteocytes were primarily located beneath the membrane, forming new bone trabeculae. Fibroblasts displayed active spindle-shaped morphology, occasionally forming chains or clusters within the porous network of the scaffold, demonstrating effective tissue ingrowth and adaptation to the nanofibrous structure.

### 3.3. Cellular Distribution in Zn-M Membranes

In Zn-M membranes, osteoblasts, M1 macrophages, fibroblasts, and blood vessels were significantly more abundant under the membrane (*p* < 0.05) (Table 1B, Figure 7 and Figure 8). Fibroblasts and vascular vessels showed similar counts at “inner” and “under” membrane locations (*p* > 0.05) (Table 1B). Osteoblasts appeared basophilic and cuboidal, often aligned along osteoid seams, in close proximity to vascular structures. Osteoclasts were observed at bone surfaces and occasionally near multinucleated giant cells. Both osteoblasts and osteoclasts appeared in close contact with marrow elements and the contiguous vasculature. Numerous osteocytes with characteristic dendritic morphology were detected within the new bone matrix (Figure 7 and Figure 8). The general cellular pattern reflected an active bone remodeling environment with intense osteoblastic and macrophagic activity.

### 3.4. Cellular Distribution in Dox-M Membranes

Dox-M presented increased cellularity and vascularization under the membrane for most cell types (*p* < 0.05) (Table 1C, Figure 9 and Figure 10). No significant differences were observed for osteoclasts and the M1/M2 ratio between locations (*p* > 0.05). Vascular vessels showed similar counts at “inner” and “under” membrane locations (P>0.05) (Table 1C). Fibroblasts and blood vessels were particularly abundant, distributed throughout the defect area and within the membrane pores, suggesting efficient cell migration and neovascularization. Histological assessment showed multiple vascular vessels and bone cells in close contact throughout the bone defect (Figure 9 and Figure 10).

### 3.5. Quantitative Comparison Among Membranes

Statistical analysis revealed distinct cellular behaviors among the three membrane types (Table 1). Zn-M exhibited the highest osteoblast counts, whereas Dox-M promoted greater fibroblast proliferation. Si-M demonstrated a balanced response with moderate osteogenic and vascular cell numbers. Osteoclast counts were lowest in the Dox-M group, while M1 macrophages were most prevalent under Zn-M membranes. Regardless of functionalization, all membranes displayed a higher density of cells and vascular structures under rather than over the membrane (*p* < 0.05).

These findings collectively demonstrate distinct biological responses depending on the type of functionalization. The implications of these results for bone regeneration and immune modulation are further addressed in the Section 4.

## 4. Discussion

In this study, the biological performance of electrospun, non-resorbable polymeric membranes characterized by controlled porosity and a high surface-to-volume ratio [41] was evaluated in vivo to assess their ability to sustain osteogenic cell activity and tissue regeneration during bone healing. Histological analyses revealed that two main cell populations—monocytes/macrophages and osteoprogenitor cells—migrated into the internal structure of the membranes. Both cell types exhibited secretory activity, releasing growth factors within the membrane matrix [22]. The highest density of cells and blood vessels was observed in the tissue underlying the membrane, compared with its inner or superficial regions. These findings confirmed the study hypothesis, providing histological and structural evidence that the membrane functions as an active biointerface rather than merely as a passive physical barrier. This supports the concept that biomimetic nanofibrous scaffolds offer more than structural support—they act as bioactive matrices capable of orchestrating both immune and vascular responses during bone regeneration. By reproducing the hierarchical architecture and biochemical cues of the native extracellular matrix, these systems demonstrate clear advantages over conventional barrier membranes used in guided bone regeneration [14,41]. In this way, the membranes themselves generated the necessary microenvironmental signals for cellular migration, adhesion, and differentiation, while promoting vascular ingrowth and the localized release of growth factors within their structure [44].

An appropriately engineered scaffold that reproduces the hierarchical architecture of cortical bone can promote both osteogenesis and endothelial-driven vasculogenesis [45]. In the study by Osorio et al. (2017), FESEM and AFM analyses revealed fiber diameters of about 300 nm and surface nanoroughness values between 82.7 and 137.1 nm [41], parameters known to enhance osteoblast attachment and proliferation [46]. Individual nanofibers tended to cluster into micro-bundles with mean diameters of 1.9–2.1 µm [41]. Adequate pore geometry and interconnectivity are essential for cell migration, nutrient diffusion, and waste removal throughout the scaffold [47]. Previous studies reported enhanced osteogenic differentiation in scaffolds with pore diameters between 5 and 8 µm [48], while Jin et al. [23] suggested an optimal range near 0.9 µm. Other authors observed that barrier membranes with pores exceeding 25 µm markedly increased early bone formation [49]. In the present membranes, inter-fiber spacing ranged from 11.5 to 110 µm [41]. Such porosity facilitates molecular transport, cellular migration, and vascular ingrowth, favoring tissue integration [47]. The overall porosity reached approximately 30% [41]. Although the optimal porosity threshold remains undefined, the present membranes meet the requirements for guided bone-regeneration applications in terms of permeability and occlusivity [47]. They also satisfy the macroporosity condition (≥100 µm) necessary for effective three-dimensional scaffolding; pores up to 110 µm were recorded [41]. The nanofibrous configuration closely mimics the physical scale of the native extracellular matrix, providing a favorable microenvironment for cellular adhesion and proliferation [18].

Newly formed woven bone was identified under the membrane, appearing either as continuous extensions from the bone margins or as isolated bone islands (Samples treated with Zn-M membranes exhibited a higher number of osteoblasts compared with the other groups. The carboxylate moieties of Zn-M likely acted as anchoring sites after the controlled release of Zn ions [50]. Osteoblasts were frequently observed near osteoclasts and marrow cells, showing their typical cuboidal morphology and alignment along newly deposited osteoid seams. The submembranous microenvironment acted as a transitional niche, supporting the migration of osteoprogenitor cells from the bone walls [6], and enabling synergistic interactions between endothelial and osteoblastic cells [30]. Such an in vivo-like environment, favoring both cell–cell and cell–matrix communication, enhanced osteogenic differentiation within the scaffold [45]. Osteoblasts predominantly localized under the membrane, where numerous vascular structures were also evident. Mature osteoblasts differentiated into osteocytes embedded in the lacuno-canalicular network of the bone matrix [51]. Osteocytes, the most abundant bone cell type, act as central regulators of bone remodeling and homeostasis [52].

Osteocytes exhibited a comparable distribution throughout the membrane, independent of the type of functionalization, although the bridging of newly formed bone typically occurred under the membrane surface [6]. Beyond regulating osteogenic differentiation, osteocytes contribute to mesenchymal stem cell (MSC) recruitment, proliferation, and osteogenic commitment through the secretion of signaling factors [45]. Considering their homogeneous distribution, it can be inferred that osteocyte function remains consistent across the entire membrane, irrespective of the histological region evaluated.

Osteoclasts were observed under and within the membrane, often associated with osteoblasts [6] and in proximity to multinucleated giant cells. The presence of these multinucleated cells did not impair bone healing, indicating that a classical foreign-body reaction could be ruled out [53]. Osteoclast-like cells were also detected along the outer surface of the membrane [21]. The lowest number of osteoclasts was recorded in the Dox-M group, suggesting reduced demineralization and enhanced bone formation due to inhibition of collagenolytic activity and promotion of matrix remineralization [54]. Doxycycline is known to suppress osteoclast activity, and inhibition of osteoclastogenesis has been shown to modulate biomaterial-mediated bone regeneration [55]. The coupling between osteoclasts and osteoblasts is a tightly regulated mechanism in which bone resorption is followed by new bone deposition, with osteoclasts orchestrating remodeling and osteoblasts subsequently secreting the osteoid matrix [24]. In addition to their resorptive function, osteoclasts can promote osteogenesis and angiogenesis through the release of bioactive molecules in their secretome. Increased osteoclast populations have been correlated with enhanced vascularization and bone regeneration in animal models of critical-size calvarial defects [55]. The amount of doxycycline loaded onto the Dox–M membranes (≈76.2 µg mg^−1^) lies within the therapeutic range reported for local antimicrobial and anti-inflammatory activity without inducing cytotoxic effects. Previous studies have shown that local Dox concentrations equivalent to 20–80 µg mg^−1^ of polymeric carrier or 20–50 µg cm^−2^ of membrane surface are sufficient to inhibit matrix metalloproteinase activity and bacterial growth, while promoting osteoblastic differentiation [11,56]. This indicates that the functionalization process achieved an adequate balance between effective drug delivery and biocompatibility, confirming the potential of these membranes as controlled-release systems for guided bone regeneration.

Overall, osteocytes, osteoblasts, and osteoclasts were rarely found within the membrane, indicating limited bone-cell ingrowth compared with the natural trabecular bone porosity, which averages around 250 µm [47]. This represents a limitation of the present study and should be addressed in future investigations using newly developed biomaterials. The few osteocytes detected may correspond to early healing stages, as only pores larger than 25 µm are typically required for initial bone ingrowth [49]. True integration requires in situ scaffold remodeling, including osteocyte incorporation, vascular infiltration, and extracellular matrix reorganization to adapt to the host site. The closer an implant replicates the structure of native bone, the faster the integration process, as local cells expend less time and energy during tissue adaptation [57].

The biological effects observed in Zn-M and Dox-M membranes can be interpreted in light of the distinct molecular mechanisms of both agents. Zinc ions are essential cofactors in numerous enzymes regulating bone formation and have been shown to enhance osteoblast differentiation through activation of the Runx2 and Wnt/β-catenin signaling pathways. In addition, Zn increases alkaline phosphatase activity, stimulates collagen synthesis, and contributes to the mineralization of the extracellular matrix. On the other hand, doxycycline exerts multiple regulatory effects on bone turnover and inflammation. It inhibits matrix metalloproteinases (MMP-2 and MMP-9), thereby reducing excessive matrix degradation and osteoclastic resorption. Furthermore, doxycycline modulates macrophage polarization, facilitating the transition from M1 (pro-inflammatory) to M2 (pro-regenerative) phenotypes, which supports angiogenesis and bone healing [11,58]. These complementary mechanisms explain the differential biological responses induced by Zn- and Dox-functionalized membranes and support their potential for targeted regenerative applications.

Fibroblasts, osteocytes, and osteoblasts contribute to the synthesis of extracellular matrix components, generating a well-organized fibrous network that establishes a specialized microenvironment for tissue regeneration. This phenomenon was particularly evident under the membrane, with Dox-M showing the highest fibroblast counts among all groups. The primary functional requirement of a GBR membrane is its barrier capacity—preventing gingival fibroblasts from invading the osteogenic compartment [23]. However, in the present study, fibroblasts were able to penetrate the membrane scaffold, proliferating not only in the pores and on the outer and lower surfaces (“over” and “under”), but also within the inner region of the membrane. Similar findings were previously described on both sides of barrier membranes by Behring et al. (2008) [59]. Morphologically, fibroblasts displayed either rounded profiles with few cytoplasmic extensions or flattened, spindle-shaped cells, the latter associated with stronger attachment to the membrane surface [59]. The presence of filopodia and lamellipodia confirmed the motile behavior of these cells, which tended to spread, align, and form cellular chains or colonies covering large scaffold areas. This organization indicates that the nanofibrous structure provided suitable conditions for cell adhesion and proliferation [60], consistent with a healthy cellular response [59]. The resulting cell arrangement supports the formation of a well-structured extracellular matrix within the polymeric membrane, serving as a cell-mediated modulatory interface for tissue regeneration [61]. These observations confirm that the electrospun fibrous membrane exhibits adequate nanoscale architecture and fulfills the functional requirements of a reliable GBR scaffold.

Vascularization of the native tissue growing into the construct, as well as differentiation of vascular-related cells, was clearly observed. Bone is inherently a highly vascularized tissue, and proper ossification depends on the simultaneous development of new blood vessels; thus, angiogenesis and osteogenesis are closely coupled processes [20]. Overall, blood vessels were more abundant within and under the membrane than on its outer surface, independent of the type of functionalization. Notably, a greater proliferation of vascular structures was detected under the membrane in the Si-M group. The incorporation of silica was intended to enhance scaffold bioactivity and osteoconductivity [62], facilitating calcium phosphate deposition on the membrane surface [10]. The vascular network established under the membrane resembled a hierarchical perfusion system containing both arterial and venous elements. Similar formation of interconnected vascular channels within silk fibroin membranes functionalized with bi-active peptides has been previously reported [28], supporting the findings of the present study. In all experimental groups, blood vessels were consistently observed under and within the membrane. It has been demonstrated that the presence of vascular structures within nanofibrous scaffolds mediates the coupling between osteogenesis and angiogenesis [63]. Foreign-body giant cells are known to secrete vascular endothelial growth factor (VEGF), a key mediator of angiogenesis during bone regeneration [22]. Coordinated interactions between osteogenic and angiogenic cells are therefore critical for organized and efficient repair. The reduced angiogenic response detected in Dox-M under the membrane may result from doxycycline-mediated inhibition of matrix metalloproteinases (MMPs) [58]. MMP-2, produced primarily by osteoblasts, plays a regulatory role in angiogenesis and bone remodeling through the VEGF/ERK1/2 signaling pathway [30]. Nevertheless, in the Dox-M group, blood vessels maintained adequate diameter and spacing, ensuring functional interconnection of the vascular network and effective tissue perfusion [28].

The marked heterogeneity of capillary vessels reflects the complexity of the local injury, inflammation, and immune responses occurring during healing [63]. Macrophages play a pivotal role in angiogenesis [24], and it is therefore likely that both macrophages and vascular structures coexist in these areas. Mononuclear cells were predominantly found under the membrane, regardless of the functionalization type, and on the upper surface in the Zn-M group. Similar findings were described by Elgali et al. (2016) [22], who reported monocytes, osteoprogenitors, and macrophages infiltrating an extracellular matrix membrane derived from porcine small intestinal submucosa, migrating through the peripheral margins of bone defects. Monocytes and macrophages are recognized as key regulators of the healing process due to their early recruitment and extensive secretory capacity. These cells sense the physicochemical properties of the implanted biomaterial and communicate this information to neighboring cells [21] by releasing a wide array of cytokines and growth factors [64]. Consequently, macrophages influence both the initial inflammatory phase and the subsequent bone regeneration outcome [24].

In the present study, M1 macrophages were predominantly located under the membrane, and their number was higher in Zn-M compared with Si-M and Dox-M groups. This pronounced M1 recruitment under the membrane-based scaffold reflects an acute inflammatory response [55] and the parallel involvement of immune cells such as T cells and monocytes [24]. The recruitment of endogenous stem cells into scaffolds implanted in vivo can be directly modulated by the local inflammatory response [55]. Dox-M showed the lowest number of M1 under the membrane, consistent with a reduced pro-inflammatory profile. Depletion of M1 macrophages leads to a significant decrease in cytokines such as IL-6, TNF-α, and interferon gamma–induced protein (IP-10) [24,65]. M1 macrophages exhibit strong bactericidal activity and depend mainly on anaerobic glycolysis and the pentose phosphate pathway to rapidly generate energy, which is essential for early infection control [24,66]. A timely transition from pro-inflammatory M1 to pro-regenerative M2 macrophages is critical for normal bone repair [55]. M2 macrophages were also observed under the membrane-based scaffold in all groups, but in Zn-M, M2 macrophages showed similar counts over and under the membrane. Moreover, M2 cells were more abundant over the membrane in Zn-M compared with Si-M and Dox-M. The number of M2 macrophages under the membrane remained relatively constant across all functionalization types. M2 macrophages are involved in tissue remodeling, repair, and wound healing, requiring continuous intracellular energy supported by oxidative phosphorylation [24,67]—a metabolic route also promoted by Zn to enhance osteogenesis [68].

To optimize the therapeutic performance of functionalized membranes, the degree of macrophage polarization must be carefully regulated, since excessive induction of either M1 or M2 phenotypes may negatively affect bone healing outcomes [24,69]. Modifying the chemical composition of the membranes could therefore serve as a strategy to modulate the M1/M2 ratio, enabling targeted immunomodulation. The dynamic switch between M1 and M2 macrophage states represents a promising direction for the design of next-generation therapeutic biomaterials [70]. Based on the present findings, specimens treated with Zn-M—showing the highest M1/M2 ratio over the membrane—might exhibit a chronic pro-inflammatory tissue response, potentially leading to unfavorable remodeling events such as fibrous encapsulation, as previously described [71]. However, the six-week evaluation period adopted in this study should be considered relatively short and constitutes a limitation of the current experimental design. Consequently, the control of macrophage polarization should remain a critical parameter in the development and testing of functionalized membranes for bone regeneration.

At present, bone tissue engineering demands the development of more efficient and clinically translatable biomaterials and scaffolds for the management of critical-size defects that fail to heal spontaneously. Future strategies for guided bone regeneration (GBR) should consider the incorporation of cells—with or without growth factors—into “smart” scaffolds designed to exhibit targeted osteoinductive and immunomodulatory properties. Additional studies are required to further elucidate the expression, secretion, and biological impact of growth factors in relation to membrane composition and structural features. One current limitation of electrospinning lies in the difficulty of generating interconnected macropores within the scaffold, which are essential for effective alveolar bone regeneration. Therefore, the integration of electrospinning with complementary fabrication techniques, such as three-dimensional (3D) printing, may represent a promising approach to producing multilayered and functionally graded surgical membranes [72]. In this context, melt electrowriting technology could serve as a bridge between solution electrospinning and 3D printing, enabling precise control of fiber orientation and scaffold architecture for improved clinical performance.

Although the developed membranes demonstrated promising biological performance, several limitations must be acknowledged. The current electrospun structure presents restricted macroporosity, which may limit deep cellular ingrowth and complete vascular penetration. The in vivo evaluation was restricted to a single healing period of six weeks, which limits the understanding of the long-term bone remodeling process and prevents assessment of late-stage mineralization and scaffold degradation. Furthermore, the rabbit calvarial model, while well established, does not fully replicate the mechanical and biological conditions of the human alveolar ridge. Another limitation concerns macrophage identification, which was based solely on morphological criteria without immunohistochemical confirmation. This approach, inherent to undecalcified histological processing, precluded molecular labeling but still allowed for consistent identification of M1 and M2 morphotypes as reported in previous studies. Future studies should explore longer healing intervals, larger animal models, and the integration of complementary manufacturing techniques such as melt electrowriting or 3D printing to enhance pore interconnectivity, validate macrophage polarization, and improve clinical translation. While quantitative histomorphometric analysis could provide additional numerical validation, the qualitative histological assessment performed in this study was considered sufficient to demonstrate the osteogenic, angiogenic, and immunomodulatory potential of the membranes. This approach ensured an ethically balanced design, minimizing animal use while achieving meaningful biological interpretation.

## 5. Conclusions

The present in vivo study demonstrates that biomimetic nanofibrous membranes functionalized with SiO_2_ nanoparticles, zinc, and doxycycline act as bioactive scaffolds capable of supporting bone and vascular tissue regeneration. The histological and quantitative findings revealed distinct biological responses depending on localization and the type of functionalization. Cellular groups were more evident under the membrane in all of the membranes studied. Zn-M promoted osteoblastic activity and angiogenesis, Dox-M modulated the inflammatory response while reducing osteoclastic activity, and Si-M provided a balanced osteogenic and vascular effect.

These results confirm the potential of multifunctional electrospun membranes to serve as controlled drug delivery systems and immunomodulatory interfaces for guided bone regeneration. From a translational standpoint, Zn-M membranes may be preferable for early osteogenic stimulation and bone defect healing, whereas Dox-M membranes may be indicated in clinical situations where modulation of inflammation and osteoclastic activity is desired, such as in periodontal or peri-implant inflammatory environments. This distinction highlights the complementary roles of each membrane type and provides a clear clinical framework for their potential application in regenerative therapies.

Future strategies in bone tissue engineering should focus on integrating such biomimetic membranes with advanced manufacturing technologies to overcome the current limitations of electrospinning, such as restricted macroporosity. Combining cellular or growth factor incorporation with tailored scaffold architectures could lead to the development of “smart” materials optimized for clinical translation.

## Figures and Tables

**Figure 1 biomimetics-10-00726-f001:**
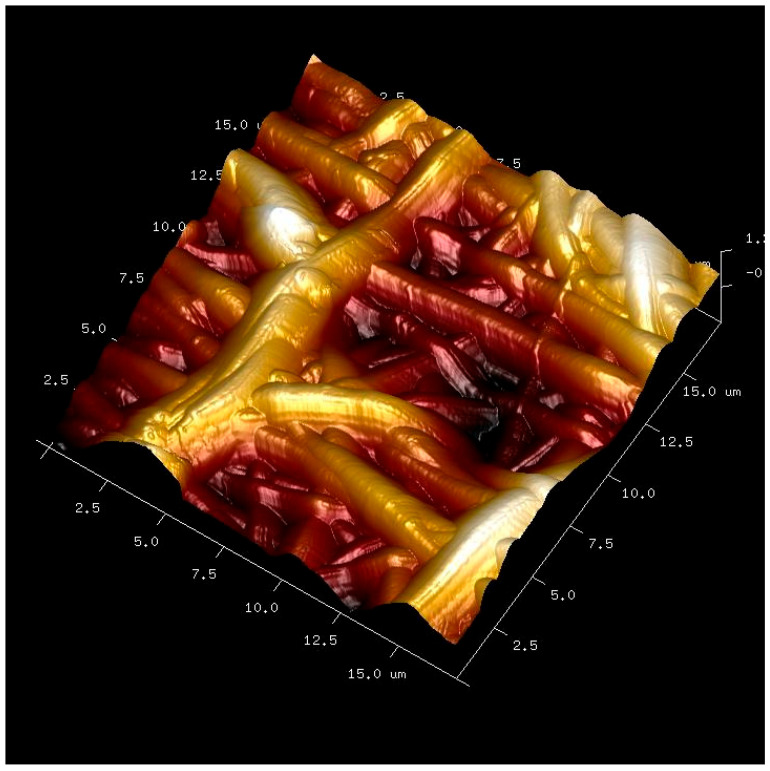
AFM image of the membrane-based scaffold (Si-M group) in guided bone regeneration. Overlapped and randomly distributed nanofibers may be observed.

**Figure 2 biomimetics-10-00726-f002:**
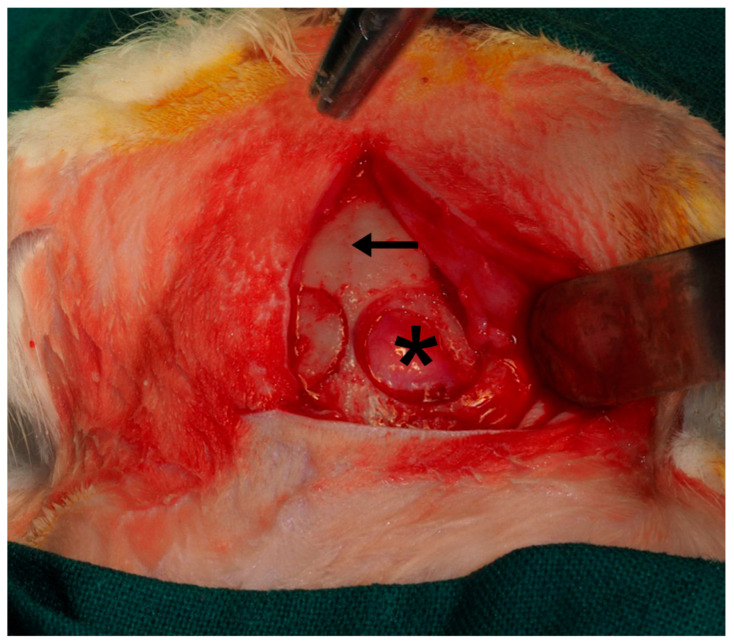
Four critical bone defects were practiced on the parietal bone of the animal. The image permits us to see the skull midline (arrow) and a complete bone defect that exhibits the *dura mater* (asterisk).

**Figure 3 biomimetics-10-00726-f003:**
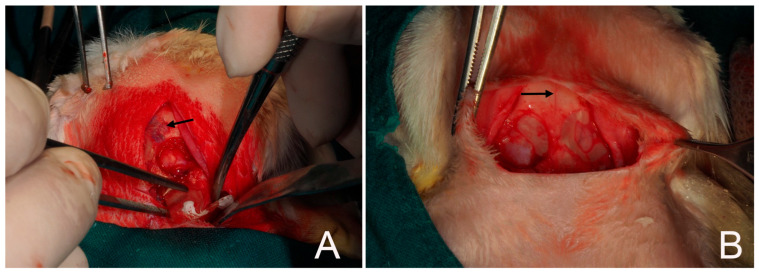
(**A**) A membrane is being placed on a calvarial defect. (**B**) Two membranes covered two of the four defects. The skull midline (arrows) is the main anatomical reference to practice the four critical bone defects.

**Figure 4 biomimetics-10-00726-f004:**
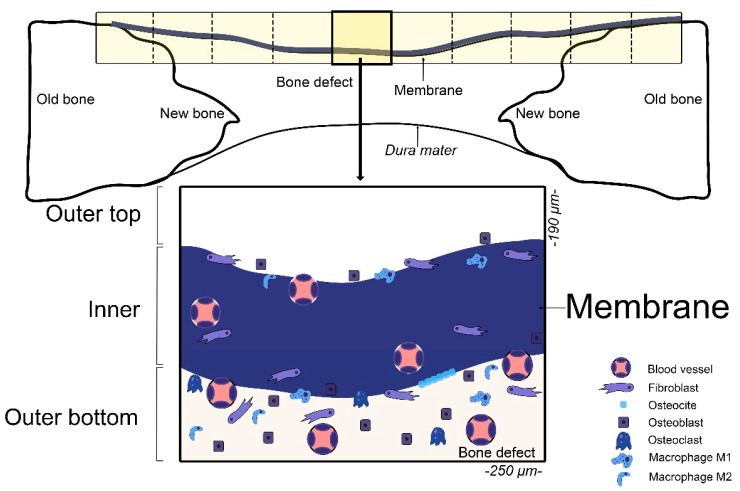
A schematic diagram showing the defect and the area of measurements for histology. A software rectangular grid consisting of multiple segmented regions (250 × 190 µm) covered the entire length of the membrane, with enough space above (outer top/over) and below (outer bottom/under) the membrane, apart the interior (inner) to analyze the histology involved. A schematic illustration from a section of the membrane and its histological environment is represented at the bottom of the image.

**Figure 5 biomimetics-10-00726-f005:**
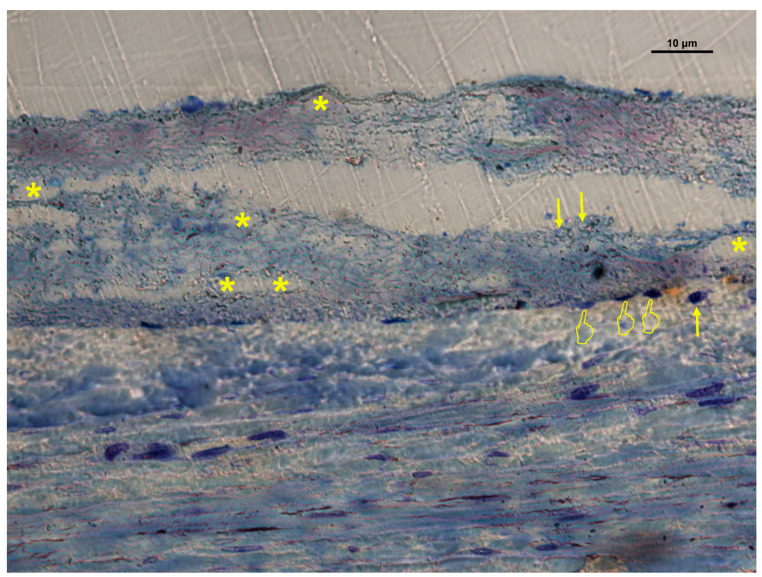
Bone histology obtained with silica-loaded membrane (Si-M), showing coloration with toluidine blue to visualize the histology features, at 6 weeks of healing time. Vascular vessels (*), osteoclasts (pointers), polymorphonuclear, foreign body giant cells and monocyte–macrophage-like cells (single arrows) are observed.

**Figure 6 biomimetics-10-00726-f006:**
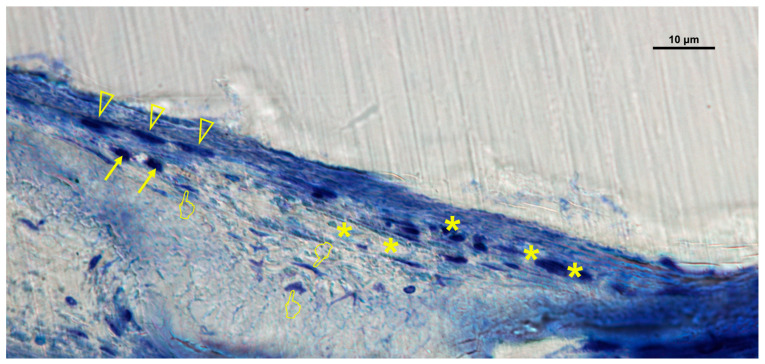
Bone histology obtained with silica-loaded membrane (Si-M), showing coloration with toluidine blue to visualize the histology features, at 6 weeks of healing time. Single arrows indicate the presence of osteoblasts. Pointers are signaling osteoclasts. Vascular vessels were identified by asterisks (*). Fibroblasts, making chains of cells, and colony layers are marked by arrowheads.

**Figure 7 biomimetics-10-00726-f007:**
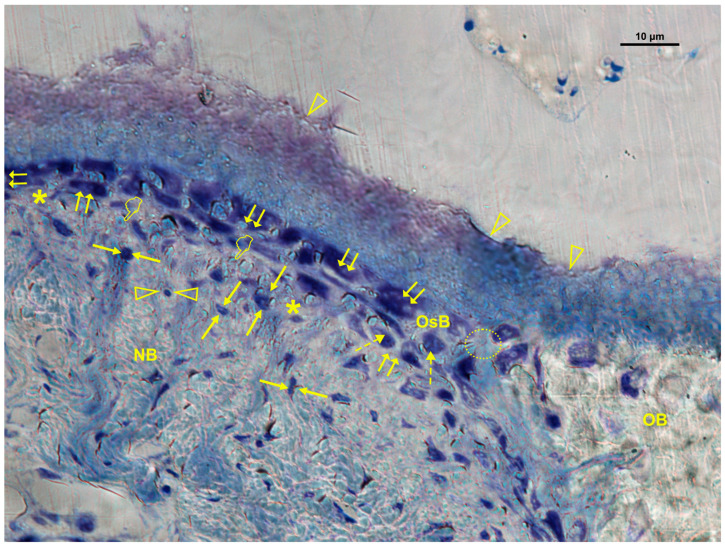
Bone histology obtained with zinc-loaded membrane (Zn-M), showing coloration with toluidine blue to visualize mineralized bone, at 6 weeks of healing time. Segments of trabecular new formed (NB) bone are observed close to the old bone (OB). Bone cells are observed in a procedure of bone remodeling. Both osteoblasts (double arrows) and osteoclasts (arrow heads) appear in close contact with marrow elements. Some entrapped osteoblasts, identified by dotted arrows, may be adverted by the osteoid bone (OsB). Interconnected contact cells (osteoblast–osteoclast) or cross-talk and extracellular matrix interactions are reflected by a dotted circle. Some blood vessels are identified by asterisks (*). Face arrows point out some macrophages. Osteocytes with their mineral lacuna (pointers) may be observed (faced arrowheads).

**Figure 8 biomimetics-10-00726-f008:**
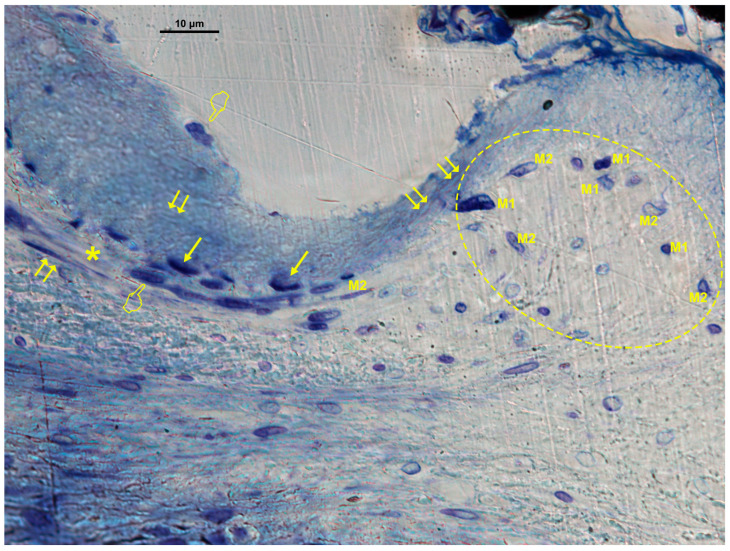
Bone histology obtained with zinc-loaded membrane (Zn-M), showing coloration with toluidine blue to visualize a canopy of macrophages and osteomacs (within the doted line), at 6 weeks of healing time. Vasculature is identified by asterisks (*). Single arrows indicate the presence of osteoblasts. Pointers are signaling osteoclasts. Fibroblasts are observed within and under the membrane (double arrows). M1, type 1 macrophages; M2, type 2 macrophages.

**Figure 9 biomimetics-10-00726-f009:**
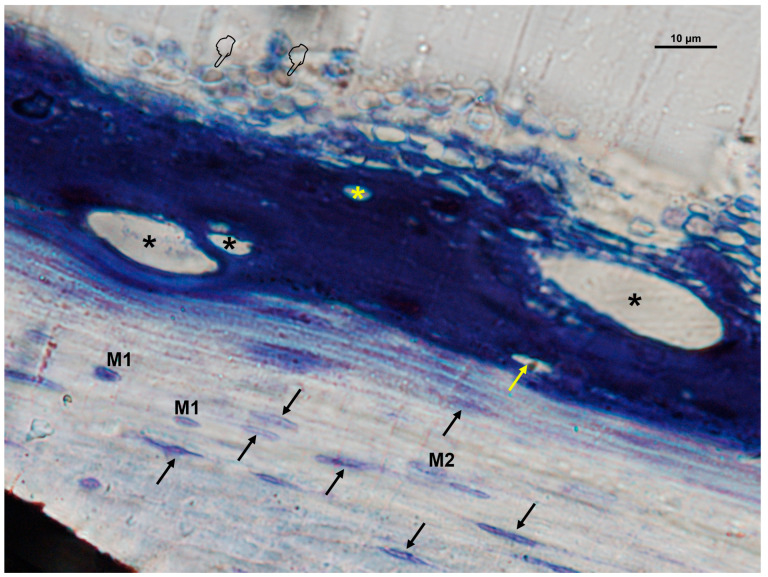
Bone histology obtained with doxycycline-loaded membrane (Dox-M), shoowing coloration with toluidine blue to visualize the different histologic components at the membrane after 6 weeks of healing time. Big profusion of vascular vessels was observed within the membrane (asterisks). Fibroblasts appear in a great number under the membrane (single arrows). M1 and M2 type macrophages are observed under the membrane. Osteocytes profess over the membrane (pointers).

**Figure 10 biomimetics-10-00726-f010:**
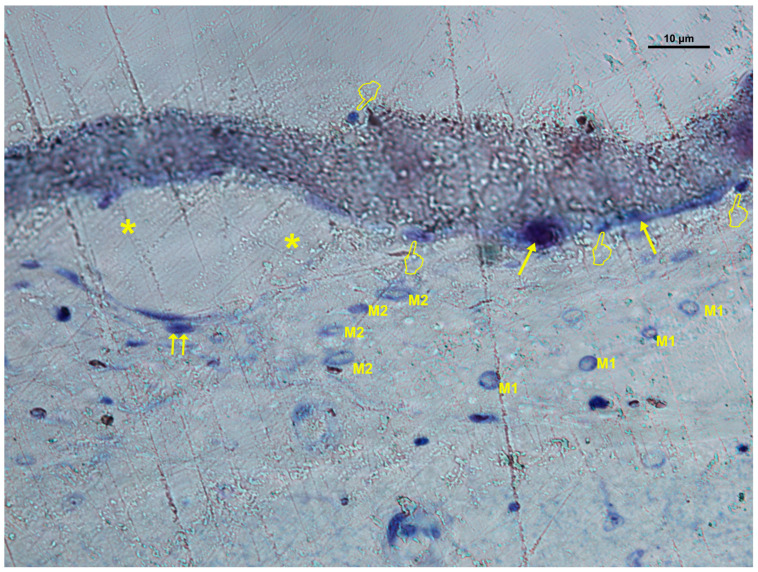
Bone histology obtained with doxycycline-loaded membrane (Dox-M), showing coloration with toluidine blue to visualize after 6 weeks of healing time the vasculature (*), osteoblasts (single arrows), osteoclasts (pointers), and a canopy of M1 and M2 macrophages. Some monocytes are also visualized (double arrows).

**Table 1 biomimetics-10-00726-t001:** Quantitative analysis of cellular and vascular components in bone defects. Mean ± Standard Error of blood vessels and bone cells counts in the selected study section over, within and under the membrane when comparing (**A**) Si-M, (**B**) Zn-M, and (**C**) Dox-M. Statistical results (*p* values) after pairwise comparisons. Similar letters indicate no differences between zones within the same cell group. Similar numbers indicate no differences between membranes within the same cell group.

				(**A**)				
	Osteocytes	Osteoblast	Osteoclast	M1	M2	M1/M2	Fibroblast	Blood Vessel
Over	0.003 ± 0.003 A1	0.324 ± 0.064 A1	0.014 ± 0.012 A1	0.600 ± 0.146 A1	0.487 ± 0.129 A1	1.134 ± 0.064 A1	0.100 ± 0.045 A1	0.00 ± 0.000 A1
Inner	0.000 ± 0.000 A1	1.072 ± 0.106 B1	0.066 ± 0.033 AB1	0.752 ± 0.114 A1	0.435 ± 0.075 A1	1.321 ± 0.078 AB1	1.517 ± 0.125 B1	0.041 ± 0.014 A1
Under	0.041 ± 0.041 A1	2.462 ± 0.180 C1	0.183 ± 0.072 B1	1.686 ± 0.190 B1	1.170 ± 0.156 B1	1.453 ± 0.102 B1	1.541 ± 0.138 B1	0.121 ± 0.026 B1
*p*	0.400	0.00	0.030	0.000	0.000	0.024	0.000	0.000
				(**B**)				
	Osteocytes	Osteoblast	Osteoclast	M1	M2	M1/M2	Fibroblast	Blood Vessel
Over	0.000 ± 0.000 A1	0.596 ± 0.120 A2	0.070 ± 0.042 A2	1.270 ± 0.218 A1	0.908 ± 0.198 A1	1.487 ± 0.142 A2	0.258 ± 0.063 A2	0.000 ± 0.000 A1
Inner	0.058 ± 0.058 A1	1.723 ± 0.183 B2	0.212 ± 0.130 A1	0.835 ± 0.176 A2	0.362 ± 0.096 B2	1.425 ± 0.114 A1	1.765 ± 0.150 B1	0.038 ± 0.14 AB1
Under	0.050 ± 0.050 A1	3.323 ± 0.227 C2	0.112 ± 0.035 A1	1.942 ± 0.256 B2	1.350 ± 0.209 C1	1.507 ± 0.128 A1	1.439 ± 0.142 B1	0.073 ± 0.021 B2
*p*	0.604	0.000	0.445	0.002	0.000	0.895	0.000	0.002
				(**C**)				
	Osteocytes	Osteoblast	Osteoclast	M1	M2	M1/M2	Fibroblast	Blood Vessel
Over	0.000 ± 0.000 A1	0.247 ± 0.051 A1	0.000 ± 0.000 A3	0.306 ± 0.089 A1	0.133 ± 0.058 A1	1.123 ± 0.046 A1	0.068 ± 0.029 A1	0.003 ± 0.003 A1
Inner	0.033 ± 0.033 A1	1.117 ± 0.114 B1	0.000 ± 0.000 A2	0.656 ± 0.123 A1	0.412 ± 0.093 A1	1.307 ± 0.083 A1	1.428 ± 0.118 B1	0.041 ± 0.012 B1
Under	0.008 ± 0.008 A1	2.491 ± 0.161 C1	0.041 ± 0.028 A2	1.344 ± 0181 B3	1.092 ± 0.157 B1	1.269 ± 0.042 A2	2.726 ± 0.199 C2	0.038 ± 0.012 B3
*p*	0.466	0.000	0.122	0.000	0.000	0.164	0.000	0.005

## Data Availability

The data presented in this study are available on request from the corresponding author.

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
