# Peer review of "Bio-Membrane-Based Nanofiber Scaffolds: Targeted and Controlled Carriers for Drug Delivery—An Experimental In Vivo Study"

_biomimetics, 2025, doi:10.3390/biomimetics10110726_

Round 1
Reviewer 1 Report
Comments and Suggestions for Authors
Starting to read this article I meat the unknown abbreviation "(MMA)1-co-(HEMA)1/(MA)3-co-(HEA)2" in the description of objects of investigation. Other abbreviation had explanation just in the first time they were mentioned. This was the reason why I started to find the sense of mentioned above abbreviation in the internet, in https://scholar.google.com/ also. Then I found the meaning in the list of Abbreviations (lines 584, 581, 583, 580) in the end of the article.
Looking for the "(MMA)1-co-(HEMA)1/(MA)3-co-(HEA)2" I got to know that the same objects of investigation were presented in articles https://www.mdpi.com/2073-4360/12/5/1201 [ref 11 - in the introduction] and https://www.mdpi.com/2073-4360/15/7/1726 [ref 45 - in Experimental part] (including Supplementary Materials https://www.mdpi.com/article/10.3390/polym15071726/s1)
Descriptions of methods of investigations, sample preparation biomimetics-3933872 were the same in all these articles.
The Abstract of article biomimetics-3933872 is about the same as in articles published before.
I think that authors made big work based on experiments in vivo and presented different aspects of results interpretation in these three article. But the connection of presented results in these three article was not demonstrated in the last article. Clear difference between the aims in these three article can hardly be found.
I would be interesting to see not only photos of tissues around implanted material after implantation but also membranes before with presentation of phisico-chemical characteristics (density, fibers dimensions). Like for example in your article https://doi.org/10.1016/j.jdent.2020.103473 No characteristics of implanted membranes were presented except the procedure of their Zn and Dox modification.
It is very much inconvenient to read comments about results (table and figures) in the end of the article, not near the photos or tables. Probably it is better to combine Results and Discussion in one part to give opportunity to read comment just after result presentation.
Part Discussions has a lot of references and looks as Introduction.
In the first sentence of Discussion authors write about "in vitro", and then in Conclusion - about "in vivo". "Six weeks after surgery, the animals were sacrificed..." So it was "in vivo".
Author Response
"Please see the attachment."

Reviewer 2 Report
Comments and Suggestions for Authors
The manuscript by Toledano et al. is devoted to studying the in vivo biological properties on a rabbit model of a poly(meth)acrylate membrane loaded with silicon oxide nanoparticles and zinc. This membrane serves as a drug-containing scaffold for local antibacterial therapy during implantation. In general, the study is properly designed, and the manuscript is well-written and illustrated. All biological results have the required statistical treatment, and the histological images have excellent labeling. However, some issues should be addressed during revision before a final decision can be made.
- In the abstract, the full name of the copolymer should be provided at the first mentioning.
- Lines 58-60. Were the mentioned materials proposed in this work or in previous studies? If the latter, a citation to the work that first reported them is required.
- In Section 2.1, it is stated that the membranes were functionalized via ionic interactions between the drug's amino groups and the matrix's carboxyl groups. To understand the effect of the immobilized drug, it is crucial to know the amount of drug adsorbed. However, this parameter was not found in the text of the manuscript. Please add it.
- In addition, the authors should discuss the drug loading level required to observe the therapeutic effect, supporting this discussion with relevant references.
- Finally, the authors should discuss the limitations of the studied material.
Author Response
"Please see the attachment."

Reviewer 3 Report
Comments and Suggestions for Authors
please find attached

Author Response
"Please see the attachment."

Round 2
Reviewer 1 Report
Comments and Suggestions for Authors
Thank you very much for your work. This version of the article is perceived much easier. I have one small question / recommendation. Is it so necessary to declare in this and in previous articles the negative "null hypothesis"? I understand that it is such a way of thinking. Probably it is my personal perception of "null" as "empty" and as negative. But probably there is a positive variant for the aim formulation. You may take this question / recommendation as a "null" comment.
Reviewer 3 Report
Comments and Suggestions for Authors
The authors significantly improved the manuscript.
Authors should remove references to tables or figures in the discussion.
